# Calcium Signals during SARS-CoV-2 Infection: Assessing the Potential of Emerging Therapies

**DOI:** 10.3390/cells11020253

**Published:** 2022-01-12

**Authors:** Sascha Berlansky, Matthias Sallinger, Herwig Grabmayr, Christina Humer, Andreas Bernhard, Marc Fahrner, Irene Frischauf

**Affiliations:** Institute of Biophysics, Johannes Kepler University Linz, Gruberstrasse 40, 4020 Linz, Austria; sascha.berlansky@jku.at (S.B.); matthias.sallinger@jku.at (M.S.); herwig.grabmayr@jku.at (H.G.); christina.humer_1@jku.at (C.H.); andreas.bernhard@jku.at (A.B.)

**Keywords:** SARS-CoV-2, calcium, viral infections, Ca^2+^ channels, therapeutics

## Abstract

Severe acute respiratory syndrome coronavirus 2 (SARS-CoV-2) is a positive-sense single-stranded RNA virus that causes coronavirus disease 2019 (COVID-19). This respiratory illness was declared a pandemic by the world health organization (WHO) in March 2020, just a few weeks after being described for the first time. Since then, global research effort has considerably increased humanity’s knowledge about both viruses and disease. It has also spawned several vaccines that have proven to be key tools in attenuating the spread of the pandemic and severity of COVID-19. However, with vaccine-related skepticism being on the rise, as well as breakthrough infections in the vaccinated population and the threat of a complete immune escape variant, alternative strategies in the fight against SARS-CoV-2 are urgently required. Calcium signals have long been known to play an essential role in infection with diverse viruses and thus constitute a promising avenue for further research on therapeutic strategies. In this review, we introduce the pivotal role of calcium signaling in viral infection cascades. Based on this, we discuss prospective calcium-related treatment targets and strategies for the cure of COVID-19 that exploit viral dependence on calcium signals.

## 1. Introduction

In late 2019, a novel, zoonotic coronavirus named severe acute respiratory syndrome coronavirus 2 (SARS-CoV-2) was described in southeastern China. Since then, SARS-CoV-2 caused immense damage to human lives and economies, being the causative organism of the coronavirus disease 2019 (COVID-19) pandemic [1,2,3,4]. Massive research effort has been put into the investigation of viral properties, leading to the development of several highly potent vaccines and giving hope to bring the pandemic phase to a close [5,6,7,8,9,10,11]. Nevertheless, the properties of SARS-CoV-2 have not yet been fully deciphered.

Since vaccines became widely available at the end of 2020, many countries have been struggling to immunize sufficient proportions of their population to contain further viral spread. Recurrent waves of infection paired with more infectious virus variants are still placing a severe strain on public health systems, preventing governments from abolishing safety regulations and returning their countries to a pre-pandemic state. Potential explanations for recurrent infection waves include mutations of the virus that lead to (partial) evasion of vaccine-derived antibodies, anti-vaccination movements spreading misinformation, unsubstantiated rejection/fear of COVID-19 vaccines, conspiracy theories as well as mistrust in pharmaceutical companies or public institutions [12,13,14,15,16,17]. Aside from countering these obstacles, effective post-infection treatments are required to finally put a stop to the overload of hospitals, particularly intensive care units (ICUs).

Calcium (Ca^2+^) as an important second messenger in excitable and non-excitable cells controls essential functions such as muscle contraction, cellular signaling processes and immune responses [18]. Intracellular and organellar calcium concentrations are tightly controlled via various pumps, ATPases, ion channels and uniporters, which have been increasingly studied over the past years. During viral infection, cellular calcium dynamics are highly affected as dysregulation of host cell signaling cascades is elicited by these infectious agents [19].

The role of calcium in virus-host cell interaction has been proven for various types of viruses, including coronaviruses. For instance, in a study from 2015, Nieto-Torres et al., discovered that the coronavirus envelope (E) protein of SARS-CoV activates the so-called NLR family pyrin domain containing 3 (NLRP3) inflammasome by functioning as a Ca^2+^ channel in endoplasmic reticulum Golgi apparatus intermediate compartment (ERGIC) as well as Golgi membranes [20]. Inflammasomes are protein complexes that oligomerize and form receptors upon the detection of pathogen-associated molecular pattern (PAMP) signals in host cells, activating caspases that produce cytokines and initiate pyroptotic cell death [21]. Importantly, the inflammatory signaling cascade that leads to the activation of the NLRP3 inflammasome, regulated by interleukin 1 beta (IL-1β), mainly depends on Ca^2+^ levels [22,23]. Due to these findings, calcium has also been considered an important factor during SARS-CoV-2 infection. Recently, Khelashvili and co-workers proposed the potential binding of calcium to six acidic residues located at the fusion protein (FP) of the SARS-CoV-2 spike protein (E819, D820, D830, D839, D843, and D848), which modulates the ability of the virus to insert into lipid membranes. Using molecular dynamics (MD) simulations they suggested that the binding of calcium enhances the interactions between the fusion peptide and the host cell membrane [24,25,26,27]. For another review on the role of Ca^2+^ during infection see [28].

Due to the importance of calcium in viral infections, several calcium entry and cellular signaling pathways are considered as feasible drug target sites. In this regard, the use of inhibitors specific for calcium ion channels was successfully proven to be effective against influenza A virus, Japanese encephalitis virus (JEV), hemorrhagic fever arenavirus (NWA), or ebolavirus [27,29,30,31,32]. Based on these results, this review addresses the potential positive effects of calcium-related treatment targets and strategies to modulate calcium entry and cellular signaling pathways on the outcome of SARS-CoV-2 infections and COVID-19.

## 2. Calcium Ion Channels and SARS-CoV-2

The pivotal role of calcium in the infection mechanisms of SARS-CoV-2 and other types of viruses warrants a detailed look at calcium ion channels. Since they are the prime mediators of calcium signaling in cells, acting in a highly spatio-temporally controlled manner by connecting different intracellular compartments and the cell interior with the extracellular milieu, it is not surprising that calcium ion channels play the most prominent roles in this regard. The following sections examine different channel types as well as auxiliary non-channel proteins and their known implications in viral infections, particularly in the SARS-CoV-2 life cycle.

### 2.1. Voltage-Gated Ca^2+^ Channels

The family of voltage-gated Ca^2+^ (Ca_V_) channels enables cells to convert changes of the transmembrane potential into intracellular, Ca^2+^-regulated processes. Ca_V_ channels are categorized into L-, T-, P/Q-, N- and R-type channels equivalent to Ca_V_1.1- Ca_V_1.4, Ca_V_3.1- Ca_V_3.3, Ca_V_2.1, Ca_V_2.2, and Ca_V_2.3, respectively [33]. Categorization of these types of channels is mainly defined via the differences in their pore-forming α1-subunit composition. The channels are predominantly found in the plasma membrane (PM) of excitable cells, as they are gated by PM depolarization via a voltage sensor and support the fastest cytosolic Ca^2+^ signals. Rapid entry of Ca^2+^ mediates a further shift of the membrane potential towards positive values, meaning that Ca_V_ channels take on essential roles in the firing of action potentials underlying pacemaker functions of neuronal or specific cardiac cells [34,35]. Relatedly, Cav channels are also indispensable for skeletal muscle contraction, force generation in the cardiovascular system, visual transduction, transcriptional regulation, neurotransmitter release as well as the secretion of hormones [35,36,37,38,39].

Voltage-gated Ca^2+^ channels have been targets for the treatment of several diseases as mutations in these channels show effects on different kinds of cell types ranging from neurological to cardiac dysfunctions. As the selective inhibition of Ca_V_ channel subtypes shows promising results to overcome a wide range of cellular dysfunctions by specifically addressing regions of the channel required for proper function, e.g., the pore-forming region, several drugs are being tested in the context of SARS-CoV-2 infections. A short discussion thereof is provided in Section 3.1, while more detailed information about Ca2v channel blockers is presented in [40] as well as [41].

### 2.2. Store-Operated Channels

First described in 1986 [42], store-operated calcium (SOC) channels in the PM provide a mechanism for raising the cytosolic calcium concentration in cells through a process termed store-operated calcium entry (SOCE). These channels open upon the depletion of the intracellular endoplasmic reticulum (ER) calcium stores via ER calcium release channels [43]. The prototypic SOC channel is known as the calcium release-activated calcium (CRAC) channel, named after the Ca^2+^-selective CRAC current (I_CRAC_) it produces [44]. The most notable features of CRAC channels are voltage independence, inward rectification including a highly positive reversal potential as well as slow activation that takes tens of seconds to minutes [43,45]. In 2005 and 2006, the key molecular components of the CRAC channel were identified: the Ca^2+^ sensor protein stromal interaction molecule (STIM), located in the ER membrane, and the pore-forming channel subunit Orai, residing in the PM [46,47,48,49,50].

Physiologically, the depletion of the ER calcium store is elicited through activation of cell-surface receptors by ligand binding. This causes activation of phospholipase C (PLC), which proceeds to catalyze the cleavage of phospholipid head groups to membrane-bound diacylglycerol (DAG) and cytosolic inositol trisphosphate (IP3). The latter subsequently binds to IP3 receptors in the ER membrane and as these function as non-selective cation channels, the opening allows Ca^2+^ to exit along the concentration gradient from the ER lumen into the cytosol. The associated dramatic decrease in the ER luminal calcium concentration [51,52,53] serves as the activation signal for the CRAC channel [54,55,56,57,58].

CRAC channels regulate intracellular Ca^2+^ concentrations in many types of cells, including T lymphocytes [59,60,61,62]. There, they are activated upon T cell receptor (TCR) stimulation, which elicits the above-described signaling cascade via PLC [59]. The significance of CRAC channels in correct T cell function is underlined by single point mutations within STIM and/or Orai that cause severe combined immunodeficiency (SCID). Affected patients’ T cells do not exhibit any detectable SOCE or CRAC channel activity [47,63,64,65]. Although T cell development usually occurs in the absence of functional CRAC channels, reduced antigen-specific proliferation, and impaired cytokine production, including interleukin-2 (IL-2), interferon-gamma (IFN-γ), and tumor necrosis factor-alpha (TNF-α) can be observed in these cells [65,66,67,68]. SCID patients are affected by both, recurrent and chronic viral infections, indicating a prominent involvement of CRAC channels in antiviral immunity mediated by cytotoxic T cells [64,65]. In the event of a viral infection, these T cells kill virus-infected cells in the acute phase and ensure long-term protection against reinfection through the generation and maintenance of memory T cells [69]. The generation, maintenance, and function of memory T cells depends on many different factors, including cytokines, T helper cells, or the frequency and strength of antigen–TCR interactions [70,71,72,73,74]. The most important task of memory T cells is their rapid expansion upon a secondary viral infection or viral reinfection, after which they differentiate into effector cells able to kill virus-infected cells to provide strong adaptive immune protection [75]. The role of CRAC channels in these processes is not completely understood, although it has been shown that SOCE is indispensable for the maintenance of virus-specific memory T cells as well as their ability to control viral infection. Specifically, STIM proteins were found to be pivotal for these processes [76]. Moreover, both STIM and Orai are recruited to the immunological synapse (i.e., the contact interface between T cell and antigen-presenting cell) during initial T cell activation with their expression levels—as well as I_CRAC_—being highly upregulated in activated T cells. The latter result provides strong evidence for a positive feedback loop where an initial TCR signal favors the expression of STIM and Orai, which then augment Ca^2+^ signaling during subsequent encounters with an antigen [77,78].

Despite their integral role in antiviral immunity, which likely also extends to SARS-CoV-2, several studies suggest that CRAC channels are involved in the inflammation-induced injury of pulmonary endothelial cells [79,80]. Additionally, CRAC channels are, as mentioned above, also required for the production of IL-2, IFN-γ, and TNF-α cytokines, which have been linked to worsened outcomes of COVID-19 [81,82,83]. A recent study conducted by Wu et al. supports the importance of Orai1 and STIM1 for SARS-CoV2 host cell infection and IFN-I signaling. Knockout of STIM1 led to strong resistance to SARS-CoV-2 infection as a result of enhanced IFN-I response whereas Orai1 knockout resulted in high susceptibility to SARS-CoV-2 infection via the downregulation of Ca^2+^ dependent antiviral transcription factors [84]. Interestingly, STIM1-knockouts and Orai1-knockouts have opposing effects on susceptibility to SARS-CoV2 infection. The discrepancy between the two different phenotypes of the STIM-KO cells compared to the Orai1-KO cells is explained by a previous paper from the same group, that showed that STIM1 KO-cells resulted in an increase in activated STING in the ER, leading to enhanced basal IFN-I secretion [85]. Based on this, trials are underway to evaluate CRAC channel inhibitors for their suitability to treat patients affected by severe COVID-19 [86] (Section 3.1 includes an in-depth discussion on this matter). It will be interesting to see whether CRAC channel inhibitors can pave the way to an effective COVID-19 treatment and thereby contribute to alleviating the strain put on healthcare systems around the globe.

### 2.3. Auxiliary Non-Channel Proteins

Apart from the aforementioned channel-forming proteins, awareness of the regulation thereof by diverse auxiliary proteins is on the rise. In the following, some modulators of the CRAC channel linked to SARS-CoV-2 are exemplarily discussed and highlighted in Figure 1.

**HSP27|**Heat shock protein 27 (HSP27) is a chaperone that interacts with a myriad of different proteins. Among the interaction partners of HSP27 is STIM1, whereby Huang et al. were able to show that a knock-down of HSP27 leads to a decrease in STIM1 expression and thereby a reduction of Ca^2+^ influx into the cell [87]. O’Brien and Sandhu discussed HSP27 vaccination as a treatment for different COVID-19 pathophysiologies. They propose potential therapeutic benefits such as attenuation of inflammation, reduced IL-1β and increased IL-10 levels, upregulation of the granulocyte-macrophage colony-stimulating factor (GM-CSF), and promotion of endothelial repair and regrowth by upregulated vascular endothelial growth factor (VEGF) [88].

**Cav-1|**Caveolin-1 is a protein that seems to have a multitude of functions in the regulation of SOC currents, from the hindrance of SOCE-activation during meiosis, to cell type-specific SOCE activation via the modulation of micro-domains associated with SOCE (for a review on this, see [89]). Depriving cells of caveolae/caveolin-1 leads to a decrease in viral uptake, viral protein expression, virion release and Cav-1 has been shown to initiate inflammatory reactions by activating leukocytes and nuclear factor-κB [90,91]. Interestingly, SARS-CoV-2 has multiple Cav-1 binding sites that might be relevant for viral transmission, whilst syncytiotrophoblasts do not express Cav-1 [92] and so far, vertical virus transmission between pregnant women and fetuses has not been observed. This led Celik et al. to suggest that a lack of caveolin expression in the syncytio-capillary barrier acts as an important factor that prevents materno-fetal transmission of SARS-CoV-2 [93].

**UNC93BI|**In a preprint of Onodi et al., interactions between different SARS-CoV-2 strains and pre-dendritic cells (pDC) are analyzed. SARS-CoV-2-induced pDC activation is shown to be dependent on IRAK4 and UNC93B1 proteins [94]. The latter is assumed to play a role in the early stages of STIM1 oligomerization [95].

**Erp57|**ERp57, a calnexin/calreticulin-associated co-chaperone [96] acts as a thiol oxidoreductase in the ER and has been shown to have a downregulating effect on SOCE via binding to STIM1 [97]. The binding of nitazoxanide to ERp57 inhibits viral replication of paramyxovirus by targeting the folding of the viral fusion protein [98] and an inhibiting effect on different coronavirus strains in vitro has been demonstrated in multiple studies [99,100,101] and specifically on SARS-CoV-2 in cell culture assays [102,103], which was reviewed in [104].

**STING|**A deficiency of STIM1 has been shown to cause activation of the Stimulator of IFN Genes (STING) [85], which activates the production of interferons (IFNs) and nuclear factor kappa B (NF-κB) in response to sensing cytosolic DNA as well as some viral RNA fragments. Berthelot and Lioté argue that polymorphisms of the *TMEM173* gene, which encodes STING, play a role in the pathogenesis of COVID-19 [105] and point out similarities between T and B cell responses to COVID-19 and STING gain-of-function models [106]. STING is activated by rising levels of cGAMP, synthesized by the cyclic-GMP-AMP synthase (cGAS). Interestingly, acetylsalicylic acid, a cGAS-inhibitor, was associated with a decrease in ICU admission as well as mechanical ventilation of COVID-19 patients [107].

**Sigma-1-receptor|**Sigma 1 receptors (σ1Rs) are located in the ER membrane and have been identified to act as SOCE modulators [108,109]. These receptors also colocalize with viral replicase proteins and interact with non-structural protein 6 (Nsp6), a SARS-CoV-2 protein. Knockout as well as knockdown of the *SIGMAR1* gene led to a reduction of SARS-CoV-2 replication [110]. For a review of σ1R ligands that could provide therapeutic benefits see Vela JM. 2020 [111].

### 2.4. Transient Receptor Potential Channels

Transient receptor potential (TRP) channels are a large family of ion channels that are commonly divided into various subgroups according to sequence homology. These are: TRPA (ankyrin), TRPM (melastatin), TRPML (mucolipin), TRPP (polycystin), TRPC (canonical) and TRPV (vanilloid) channels [112]. TRP channels respond to temperature, mechanical stimuli, Ca^2+^ store depletion as well as chemical signals or show constitutive activity and are regulated by lipids [112,113]. TRP channels reside in the PM as well as in the membranes of mitochondria and the ER [113]. The subgroups also differ in terms of ion selectivity, e.g., TRPM channels primarily select for monovalent cations while especially the TRPV channels TRPV5 and TRPV6 channels exhibit selectivity for Ca^2+^ [112]. Thus, selected TRP channel members also act as mediators of intracellular Ca^2+^ homeostasis [113,114].

With regard to SARS-CoV-2, it has been found that TRP channels are expressed in tissues that are frequently attacked by the virus during infection (Figure 1). Indeed, TRPML2 channels were recently shown to be involved in viral entry/endocytosis of SARS-CoV-2 into host cells. The channels thereby increase the efficiency of viral trafficking within the endosomal system [115,116,117]. Moreover, a significant amount of COVID-19-related deaths is caused by edema in the lungs that leads to acute respiratory distress syndrome (ARDS) [118,119]. In this context, there is evidence that implicates TRPV4 and TRPC6 channels in pulmonary edema as well as TRPV4 and TRPM7 channels in pulmonary fibrosis [120,121,122,123]. TRP channels are also linked to the development of a variety of COVID-19 symptoms and systemic consequences, including fever, inflammatory response, neurological alterations, myalgia, headache as well as cardiovascular and gastrointestinal complications [117]. Taken together, recent scientific evidence suggests that TRP channel members are potential targets for interfering with the life cycle of SARS-CoV-2.

TRP channels are strong contributors to Ca^2+^ signals during viral infection. Their involvement in intracellular Ca^2+^ trafficking likely means that the channels play a role in establishing a favorable environment for viruses [124,125,126]. Blockage, inhibition, and/or modulation of select TRP channels could thus prove to be effective means of therapy that are not limited to SARS-CoV-2 and COVID-19. Indeed, several substances are already in clinical trials and the use of phytochemicals and medicinal plants is also being evaluated [117,127,128,129] (Section 3.2 discusses this in more detail). Due to the many existing links between TRP channels and COVID-19, these channels must be seriously considered as targets for treatment as well as prevention. Further research in this direction is indeed required.

## 3. Emerging Treatment Strategies

Since the first verified case of COVID-19 infection was reported in December 2019 in Wuhan, scientists, doctors, and pharmaceutical companies all over the world have focused on the development of therapeutics against the causative pathogen SARS-CoV-2. Although vaccines based on mRNA or adenovirus technology have been approved in many countries, there is still a need for safe and effective antivirals for the treatment of acute severe or mild SARS-CoV-2 infections. Currently, an RNA-dependent RNA polymerase inhibitor named Remdesivir that was initially developed for Ebola treatment is used for the treatment of SARS-CoV-2 patients and shows improvement in symptoms in 68% of treated patients in clinical trials [130,131]. Aside from Remdesivir, different target sites for COVID-19 drug development are currently evaluated. Molnupiravir, a prodrug of the nucleoside derivative N4-hydroxycytidine, was developed as an anti-influenza drug and acts as an inhibitor of viral RNA replication by the induction of copying errors during replication [132,133,134]. Paxlovid is a prospective oral drug combination of PF-07321332 and Ritonavir, which blocks the activity of the SARS-CoV-2-3CL protease [135,136].

The regulation of calcium ions in different cell types could be an additional potent target. As mentioned previously, Ca^2+^ plays important roles in viral entry, gene expression and was linked to imbalanced Ca^2+^ homeostasis during viral infection [19,20,22,23,27,29,30,31,32]. The inhibitory effect of calcium channel blockers (CCB) during or pre-viral infections has been reported for various types of viruses and is therefore also assumed to be a potential target site for the treatment of COVID-19 infections. Verapamil, an influenza A virus inhibitor, hydrochloride, cilnidipine and manidipine, inhibitors for Japanese encephalitis virus (JEV), and the L-type Ca_V_ channel inhibitor gabapentin effective against hemorrhagic fever arenavirus (NWA) are just a few examples for reported successful CCB use after or pre-viral infections [29,30,31].

### 3.1. Voltage-Gated Channels

A panel of voltage-activated L-type CCBs including amlodipine, nifedipine, felodipine, verapamil and diltiazem, which are primarily used to treat cardiovascular diseases including hypertension, were selected as potential drugs to inhibit SARS-CoV-2 infection in cell culture experiments using epithelial kidney (Vero E6) and epithelial lung (Calu-3) cells. All these chemical compounds bind to the α_1_ subunit of L-type Ca_V_ channels, potentially inhibiting host cell entry (Figure 1). Straus et al., suggested felodipine and nifedipine as most promising candidates against SARS-CoV-2 infection due to their high selectivity, low cytotoxicity, and their high selectivity index (SI) score but they also stated it is currently unclear how the efficacious doses they used in their study could be translated into clinical use [137].

An additional study by Solaimanzadeh also suggested that nifedipine and amlodipine reduce mortality and decrease the risk of intubation for elderly patients hospitalized with COVID-19. This effect could be induced by relaxation of pulmonary smooth muscles causing pulmonary vasodilation and therefore improve the hypoxia conditions of patients under the respective treatment [138].

A retrospective clinical report of COVID-19 patients treated with amlodipine besylate was associated with a reduced fatality rate of patients with hypertension by significantly inhibiting the post-entry replication events of SARS-CoV-2 in vitro, as reported by Zhang et al. A combined treatment with chloroquine and amlodipine besylate further enhanced the inhibitory effect in Vero E6 cells [139].

Another retrospective cohort study conducted by Peng et al. investigated the potential positive effect of CCBs particularly verapamil, on patients suffering hypertension, which might be an independent risk factor during COVID-19 infection [140,141]. In this study, they demonstrated via propensity score-matching analysis, that CCB treatment is associated with lower all-cause mortality of COVID-19 patients with hypertension. They did not observe any effects of renin-angiotensin-aldosterone system (RAAS) inhibitors or beta-blockers on patients infected with SARS-CoV-2 [141].

Nifedipine was shown to have an anti-inflammatory effect by suppressing the production of IL-1α, IL-6, and IFN-γ from peripheral blood mononuclear cells [142]. IL-6 and IFN-γ, which are known as mediators of cytokine storm in patients suffering COVID-19, could therefore serve as potential target sites by nifedipine treatment. For more detailed information about cytokine storms in COVID-19 see [143].

A rather unspecific CCB study by Mahgoub et al. reported some bioisosteres of pyrimidines as novel CCBs with a potential angiotensin-converting enzyme 2 (ACE2) inhibitory effect (Figure 1). The authors tested different chemical compounds and observed a decrease in the tone of ileal contractions in a dose-dependent manner, similar to the relaxation of K^+^-induced contractions caused by the standard drug nifedipine. They stated that the observed spasmolytic effect might be mediated through Ca^2+^ channel inhibition. Therefore, several potential calcium channel blockers together with ACE2 inhibitors, which suppress ACE2 binding to SARS-CoV-2 spike receptor-binding domain (RBD), and potential anti-inflammatory activity, mediate the reduction of IL-6 and C-reactive protein (CRP) production in LPS-stimulated THP-1 cells. The properties of the mentioned agents could be used as therapeutic tools for hypersensitive patients against COVID-19 infection and anti-inflammatory activities [144].

Yang et al., identified neferine as a possible coronavirus entry inhibitor out of 89 plant-derived natural small molecules. They reported in in vitro studies that neferine effectively protects around 75% of HEK293/hACE2 and HuH7 cells from infection by different coronavirus pseudovirus particles (SARSpp) by inhibiting host cell Ca^2+^ channels, leading to inhibition of membrane fusion and suppression of virus entry. However, which calcium channels are specifically inhibited by neferine is not defined more precisely, as the studied cell types do express different types of calcium channels. The important role of calcium in SARS-CoV-2 entry was also reported in this study, showing that a specific chelator named BAPTA-AM significantly suppressed SARS-CoV-2 pseudovirus particle infection. Follow-up experiments in this study also showed that neferine could inhibit not only early SARS-CoV-2 pseudovirus particle infection, it was also effective against SARS-CoV-2 variants as well as two other highly pathogenic human coronaviruses, namely SARS-CoV and MERS-CoV [145].

### 3.2. Store-Operated Channels

Besides the use of blockers specific for L-type Ca_V_ channels, additional calcium entry pathways could also be an interesting target for antiviral drugs against COVID-19 infection. Therefore, companies such as Calcimedica specifically address the CRAC channel pathway as a potential drug target. Previous studies have shown that CRAC channels are involved in the inflammation-induced injury of pulmonary endothelial cells, leading to loss of alveolar-capillary barrier function and extravasation of fluid into the alveoli [79,80,81,146]. Proinflammatory cytokine storms induced via CRAC channel activation may contribute to severe COVID-19 progression [79,80,81,82,83]. To prevent such arising complications, CRAC channel inhibition may have potential therapeutic benefit for patients suffering from SARS-CoV-2 (Figure 1). For detailed information about blockers used for CRAC channel systems, please refer to reviews [147,148].

In a phase 2, open-label, randomized, multicenter study of patients with severe or critical COVID-19 pneumonia by Miller et al., a novel intravenously administered nanoemulsion formulation of CM4620—named Auxora—was tested as a potential therapeutic substance that specifically inhibits the CRAC channel system [86]. CM4620, which shares structural similarities with Synta66, another specific CRAC channel inhibitor, is also undergoing phase 2 clinical trials for moderate to severe acute pancreatitis. It has been shown that under acute inflammatory conditions, this compound acts on reducing inflammatory signals in the lung, protecting tissues from calcium-induced damage as well as lowering serum and pulmonary proinflammatory cytokine levels by inhibiting the CRAC channel system [79,80,83,149,150,151] (https://clinicaltrials.gov/ct2/show/NCT03709342 (accessed on 11 November 2021)). The study by Miller et al. involved 30 persons and showed a seven day shorter median recovery time of patients with severe COVID-19 pneumonia compared to patients treated with standard of care and showed that only 18% of persons treated with Auxora needed intubation (50% of patients treated with standard of care were intubated). They also reported that the baseline P_a_O_2_/F_i_O_2_ of patients is linked to improved clinical outcomes of COVID-19 infection. Medication with Auxora showed the most promising results for patients with a baseline P_a_O_2_/F_i_O_2_ between 101–200 mmHg whereas no patients receiving Auxora or standard of care treatment with a baseline P_a_O_2_/F_i_O_2_ of >200 mmHg required invasive mechanical ventilation. A statistically significant difference in clinical improvement, indicated by the mean of an 8-point ordinal scale, was observed for patients treated with Auxora in comparison to standard of care after six days. The drug rapidly distributes to the lungs, where it is able to specifically inhibit CRAC channel-dependent cytokine release and it is stated to be reversible within 24 to 48 h [86]. Miller and co-workers provide a possible therapeutic method with the specific CRAC-channel inhibitor Auxora for patients suffering severe COVID-19 pneumonia, with direct effects on the pulmonary endothelium and indirect effects on proinflammatory cytokine production, but they also state that the sample size of patients in this study needs to be improved and a larger, double-blind, placebo-controlled study needs to be conducted to get further insights into the mode of action and impact of Auxora [86].

### 3.3. Transient Receptor Potential Channels

As already briefly mentioned in Section 2.4, there are several strategies to block, inhibit and/or modulate TRP channels currently in development. GSK2798745 is a TRPV4 antagonist that was developed to treat pulmonary edema associated with heart failure, while another substance inhibiting TRPC4 and TRPC5 targets depression and anxiety disorder [127,128]. In addition, TRP channels are also affected by various venomous toxins [152,153,154,155,156,157] which have been suggested to be used for the treatment of COVID-19, most notably resiniferatoxin (RTX) [158]. It will be interesting to see which of these candidates have the potential to be applied for COVID-19 in a widespread clinical setting.

Plants have a long history of being rich sources of active ingredients that are helpful against various kinds of disease, including viral infections. Accordingly, a recent study has shown that administration of *Nigella sativa* L. (black cumin) over 2 weeks reduces the severity of COVID-19 symptoms and elicited viral clearance [159]. The active compound in *N. sativa* seems to be thymoquinone, which shows high efficacy against SARS-CoV-2 according to recently published findings [160,161]. The link to TRP channels is provided by yet another study, which finds that *N. sativa* also affects the expression of TRP channels [162]. The non-psychotropic phytocannabinoid cannabidiol (CBD) also is a promising candidate compound. Wang and co-workers cultivated hundreds of different *Cannabis sativa* L. strains and found that those high in CBD decrease expression of ACE2 and transmembrane protease, serine 2 (TMPRSS2), which are both required for SARS-CoV-2 infection (Figure 1) [163,164]. Several TRP channels are thought to act as CBD receptors, which may explain the mechanism of action of this compound [165]. Aside from this, there are many other natural substances derived from plants that have the potential for SARS-CoV-2 inhibition, such as berbamine, resveratrol, quercetin, curcumin, spermidine, spermine, baicalin, and naringenin. Finally, dietary constituents also could play a role in the inhibition of SARS-CoV-2, as spices such as allicin, capsaicin, curcumin, gingerol, piperine, and wasabi interact with TRP channels [117,166,167]. Clearly, further research is needed to evaluate the usefulness of all these candidate substances. In any case, TRP channels seem to be the primary targets for many of them [117,168].

## 4. Discussion and Conclusions

Besides the state-of-the-art COVID-19 treatment drug Remdesivir, and potential antiviral drugs in different stages of testing such as Molnupiravir and Paxlovid, calcium channel blockers might be beneficial for treatment by modulating different pathological pathways of infected cells. The role of calcium during viral infection and especially in SARS-CoV-2 is increasingly studied but not yet fully understood. Therefore, the inhibition of calcium transport across membranes and inside cells is an interesting target site that may affect SARS-CoV-2 infection and can have positive effects on severe courses of COVID-19. Several specific, previously known, and newly developed calcium channel blockers target different channel types and host cell or viral compartments/pathways. A hypothetical mode of action using CCB treatment is the reduction of intracellular calcium levels, which would affect the viral calcium manipulation of host cells and inhibit viral host cell entry [137,144,145,163,164], has potential anti-inflammatory activity [86,142,144], and leads to relaxation of pulmonary smooth muscles, causing vasodilatory effects in the lungs and vascular system [138]. As knockout of STIM1, which results in increased STING activation, leads to enhanced IFN-I responses shown by Wu et al., the use of CCBs could possibly alter IFN-I levels and therefore also have positive effects on the resistance to SARS-CoV-2 infection. [84,85] Despite the observed positive effects of different CCB drugs/small molecules linked to COVID-19, none of them are currently in clinical use and only Auxora from Calcimedica completed Phase 2 clinical trials for treating acute pancreatitis (https://clinicaltrials.gov/ct2/show/NCT03709342 accessed on 11 November 2021). The latter is also considered to possibly have a beneficial effect for COVID-19 patients (https://clinicaltrials.gov/ct2/show/NCT04345614 (accessed on 11 November 2021)). The question as to how CCBs can be used for the treatment of viral diseases such as COVID-19 also strongly depends on the affected cell types. Since CRAC channels are widely expressed across different cell types, targeting these channels might have drastic consequences on the whole organism. The ideal timing, duration, quantitative drug management, and dose of CCBs such as Auxora are currently being tested and evaluated, whereas it is unlikely that complete inhibition of, e.g., CRAC channels is necessary to achieve positive effects as it was shown for other drug targets in the past. Due to the different functions of calcium before and during viral infections and depending on the severity of the disease, it is necessary to optimize the dose and timing of treatment—either early or late administration with CCBs to specifically target the viral life cycle, decrease viral load and to avoid a cytokine storm. Further research is needed to gain a better understanding of how and when CCBs should be used during/before viral and especially COVID-19 infections. In addition to the inhibition of specific channels mentioned in this review, side effects, such as depolarization of membrane potentials and the inhibition of CRAC channel-driven cytokine release from T cells through blocking of K^+^ channels or Cl^−^ channels need to be considered and could lead to beneficial outcomes of diverse viral infections (for detailed information, see commentary/reviews [169,170,171]). Apart from the inhibition of cellular calcium, hypocalcemia, which is defined as serum levels of ionic Ca^2+^ being below 1.18 mmol/L threshold, was reported in a number of viral diseases and also in patients suffering from COVID-19 [172,173,174,175]. Since the significance of hypocalcemia is not yet fully understood, i.e., whether it has protective or harmful effects on COVID-19 patients, clinical trials investigating the effects of calcium supplementation need to be carried out [172,176]. Despite the uncertain role of calcium supplementation, various studies reported a beneficial effect of adequate supplementation of vitamins and micronutrients on SARS-CoV-2 infection outcomes (for a detailed review on this topic, see [177]). To the authors’ knowledge, no drugs that target the auxiliary proteins described in the parts above that regulate CRAC activities are available and no mutations or polymorphisms of calcium channels or auxiliary proteins that affect host susceptibility to SARS-CoV-2 are known so far. Overall, given the multitude of indications that CCBs may contribute to a better outcome of COVID-19 disease, it might be worthwhile to focus research in this direction while keeping an eye on the future endemic state of SARS-CoV-2.

## Figures and Tables

**Figure 1 cells-11-00253-f001:**
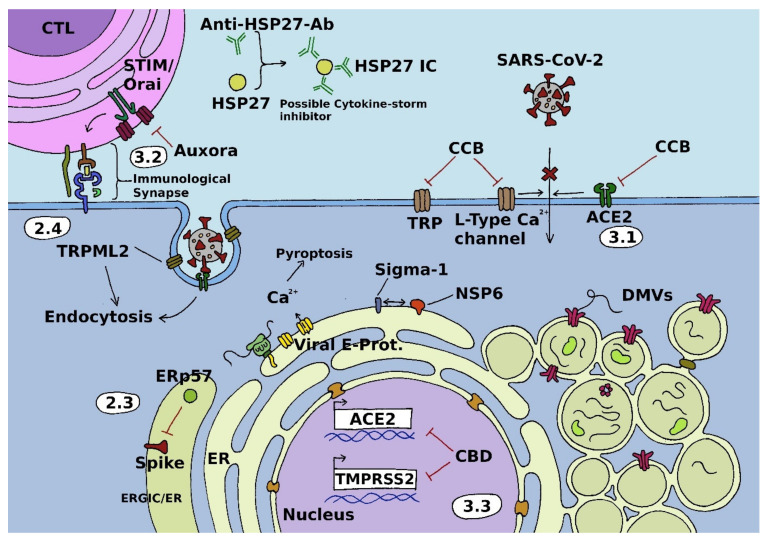
Strategies for inhibition of cellular calcium transport systems involved in viral infection cascades. Calcium levels dictate the activity of the NLRP3 inflammasome, eventually leading to pyroptosis of the cell. CCBs have been shown to interact with TRP, L-type Ca^2+^ channels as well as ACE2 and thus have the potential to inhibit SARS-CoV-2 endocytosis. Auxora is a specific inhibitor of the STIM/Orai system, which has been implicated in inflammation-induced injury of pulmonary endothelial cells as well as proinflammatory cytokine storms that contribute to severe COVID-19 disease. There is evidence that TRPML2 channels have a role in the endocytosis of SARS-CoV-2 into host cells. Potential drugs designed for these channels could be strong tools in the treatment of COVID-19. CBD decreases expression of ACE2 and TMPRSS2, which are both integral players in SARS-CoV-2 infection. Double-membrane vesicles (DMVs) are derived from ER membranes and formed by non-structural proteins (NSPs) 3–6 to facilitate viral RNA replication in separate compartments, as it would be impaired by immune-responses in the cytosol. HSP27 vaccination might attenuate inflammation and increase tissue regeneration. Inhibition of ERp57 might impair correct spike-folding during virus replication. Knockout/knockdown of the Sigma-1 receptor has been shown to reduce viral replication, Numbers in bubbles denote sections describing the indicated proteins and mechanisms in detail.

## Data Availability

Not applicable.

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
