# Peer review of "Calcium Signals during SARS-CoV-2 Infection: Assessing the Potential of Emerging Therapies"

_cells, 2022, doi:10.3390/cells11020253_

Round 1

Reviewer 1 Report

This review by Sascha Berlansky and colleagues describes evidence that targeting calcium signalling, particularly plasma membrane calcium channels, could be an effective and novel approach to treat lung disease that develops in patients with COVID-19. As expected from Irene Frischauf's group, the review is informative, well-written, comprehensive and crystallises a novel concept that I find thoughtful and convincing. In short, this is an excellent article that will be of considerable interest to the a readership well beyond the calcium signalling community. The authors are to be congratulated on producing such a rigorous and exciting piece of work. 

I have a few minor comments suggestions.

  1. In the authors' discussion of the Wu et al paper (Journal of Immunology 2021), the reader might be confused by the observation that KO of STIM1 or Orai1, which are the two essential components of the CRAC channel, have opposing effects on susceptibility to SARS-CoV2 infection. One would have expected the same phenotype in both cases. The answer lies in a previous study from the group that showed KO of STIM1 resulted in an increased presence  of STING in the ER, leading to enhanced basal IFN-gamma secretion. Perhaps this should be added.
  2. The neferine study (Yang et al) is confusing. That paper claimed neferine blocked 'host calcium channels' (whatever that means) and this resulted in inhibition of coronavirus entry. These experiments were done on HEK293 cells expressing ACE2. It is not clear what the calcium channel is. I assume it is a SOC but Yang et al cite the following paper (https://pubmed.ncbi.nlm.nih.gov/12466045/) which reports neferine blocks voltage-gated Na+, K+ and Ca2+ channels. Since HEK293 cells do not express Voltage-gated calcium channels, the target of the drug is unclear. I think the authors should discuss this point.
  3. Line 36- in addition to the reasons the authors put forward, I would suggest adding that the virus mutates, meaning it might partially evade vaccine-derived antibodies.
  4. Line 200- depriving not depraving. 

Author Response

We thank the reviewer for her/his positive evaluation of our manuscript and helpful suggestions.

I have a few minor comments suggestions.

  1. In the authors' discussion of the Wu et al paper (Journal of Immunology 2021), the reader might be confused by the observation that KO of STIM1 or Orai1, which are the two essential components of the CRAC channel, have opposing effects on susceptibility to SARS-CoV2 infection. One would have expected the same phenotype in both cases. The answer lies in a previous study from the group that showed KO of STIM1 resulted in an increased presence  of STING in the ER, leading to enhanced basal IFN-gamma secretion. Perhaps this should be added.

As suggested, a discussion of this matter is now included in the manuscript (lines 166-171).

  1. The neferine study (Yang et al) is confusing. That paper claimed neferine blocked 'host calcium channels' (whatever that means) and this resulted in inhibition of coronavirus entry. These experiments were done on HEK293 cells expressing ACE2. It is not clear what the calcium channel is. I assume it is a SOC but Yang et al cite the following paper (https://pubmed.ncbi.nlm.nih.gov/12466045/) which reports neferine blocks voltage-gated Na+, K+ and Ca2+ channels. Since HEK293 cells do not express Voltage-gated calcium channels, the target of the drug is unclear. I think the authors should discuss this point.

We now address this point in the paragraph describing the study by Yang and co-workers (lines 352-354).

  1. Line 36- in addition to the reasons the authors put forward, I would suggest adding that the virus mutates, meaning it might partially evade vaccine-derived antibodies.

Evasion of vaccine-derived antibodies by viral mutation has been added as an additional reason.

  1. Line 200- depriving not depraving. 

Corrected (now line 207).

Reviewer 2 Report

Berlansky S. et al; “Calcium signals during SARS-CoV-2 infection: Assessing the potential of emerging therapies”.

This review provides a good overview of Ca2+ signaling in viral infection cascades, including the SARS-CoV-2 virus and as such is timely due to the ongoing COVID-19 pandemic. Overall, it lists the literature on blockers of various types of Ca2+ channels and their known (if any) effect on SARS-CoV-2 pathogenesis. While it is a good bird’s eye overview, the review can benefit from some detailed explanation and appropriate citations as detailed below:

  1. While discussing SOC channels in section 2.2, the authors should cite all the papers describing identification of STIM/ORAI proteins and not just reference # 46 and 47.
  2. The authors should ideally include a schematic for activation of CRAC channels and known mechanisms of how SOCE affects SARS-CoV-2 infection/propagation. Also discuss how blocking CRAC channels can affect host susceptibility to SARS-CoV-2 by altering baseline IFN-I levels.
  3. Since Ca2+ entry can play a dual role in viral infections in terms of being necessary for viral infections (necessary for viral entry, viral exit) or deleterious to host cells by enhancing cytokine storms, the authors should discuss how blockers of Ca2+ channels can act at different stages of virus infections – early in the infection v/s later on during hospitalization.
  4. Are there any known mutations/polymorphisms in known Ca2+ channels that affect host susceptibility to SARS-CoV-2? This aspect should also be discussed.

Minor comments:

The review can benefit from editing for spell checks.

Author Response

We thank the reviewer for her/his positive evaluation of our manuscript and helpful suggestions.

  1. While discussing SOC channels in section 2.2, the authors should cite all the papers describing identification of STIM/ORAI proteins and not just reference # 46 and 47.

The STIM/ORAI papers suggested by the reviewer have been cited in section 2.2 (references 48-50).

  1. The authors should ideally include a schematic for activation of CRAC channels and known mechanisms of how SOCE affects SARS-CoV-2 infection/propagation. Also discuss how blocking CRAC channels can affect host susceptibility to SARS-CoV-2 by altering baseline IFN-I levels.

As the present review is focused on the overall importance of calcium signaling in SARS-CoV-2 infection, we feel that a special focus on CRAC channels is beyond the scope of this manuscript. Such in-depth descriptions would be more suitable in the context of a detailed CRAC channel paper. Furthermore, evidence if and how SOCE affects SARS-CoV-2 infection/propagation is still largely lacking.

Possible effects of CCB-elicited changes in IFN-I levels on host susceptibility to SARS-CoV-2 are now mentioned (lines 445-448).

  1. Since Ca2+ entry can play a dual role in viral infections in terms of being necessary for viral infections (necessary for viral entry, viral exit) or deleterious to host cells by enhancing cytokine storms, the authors should discuss how blockers of Ca2+ channels can act at different stages of virus infections – early in the infection v/s later on during hospitalization.

This point is now discussed in the manuscript (lines 460-465).

  1. Are there any known mutations/polymorphisms in known Ca2+ channels that affect host susceptibility to SARS-CoV-2? This aspect should also be discussed.

To our knowledge, no such mutations/polymorphisms of calcium channels or auxiliary proteins are known to date. A corresponding statement has been added to the discussion and conclusion section (lines 478-481).

Minor comments:

The review can benefit from editing for spell checks.

The manuscript has been carefully checked for spelling and other errors by a native speaker.